# Risk Factors Related to Resting Metabolic Rate-Related *DNAJC6* Gene Variation in Children with Overweight/Obesity: 3-Year Panel Study

**DOI:** 10.3390/nu16244423

**Published:** 2024-12-23

**Authors:** Jieun Shin, Inhae Kang, Myoungsook Lee

**Affiliations:** 1Department of Biomedical Informatics & Healthcare Data Science Center, College of Medicine, Konyang University, Daejeon 35365, Republic of Korea; jeshin@konyang.ac.kr; 2Department of Food Science and Nutrition, Jeju National University, Jeju 63243, Republic of Korea; inhaek@jejunu.ac.kr; 3Department of Food & Nutrition & Research Institute of Obesity Sciences, Sungshin Women’s University, Dobongro-76gagil-55, Kangbuk-ku, Seoul 01133, Republic of Korea

**Keywords:** three-year panel study, children obesity, resting metabolic rate (RMR), *DNAJC6*, energy expenditure, dietary fat, dietary retinol

## Abstract

This study investigated how the *DNACJ6* gene variation related to RMR alteration affects risk factors of obese environments in children with obesity aged 8–9. Methods: Over a three-year follow-up period, 63.3% of original students participated. Changes in the variables (anthropometrics, blood biochemistry, and dietary intakes) were analyzed and compared between those without obesity (non-OB) and with obesity (OB) classified at the study endpoint. Result: The average MAF of nine SNPs (*D-1* to *D-IX*) was defined as 18.1%. The OB group showed greater increases in RMR, BMI, WC, and SBP, while the non-OB group had significantly greater increases in HDL and intakes of nutrients (e.g., total calories, vitamins B2, C, folate, A, retinol, iron, and zinc). Increased RMR, BMI, BW, and RMR/BW changes were observed with mutant allele of *D-I* SNP, which was also associated with a higher prevalence of obesity. Greater increases in animal fat intake, including saturated fatty acids and retinol, were noted in the minor alleles of *D-VI, D-VII*, *D-VIII*, and *D-IX* SNPs compared to those of the major alleles. The odds ratio for BMI risk was significantly higher in the mutant alleles of *D-I (rs17127601), D-VII (rs1334880),* and *D-VIII (rs7354899)* compared to the wild type, with increases of 2.59 times (CI; 1.068–6.274), 1.86 times (CI; 1.012–3.422), and 1.85 times (CI; 1.008–3.416), respectively. RMR was a mild risk factor in minors of the *D-1, D-VII, and D-VIII*; however, a higher RMR/BW ratio significantly correlated with decreased BMI risk, and this effect was found in only the major alleles of *D-I, D-VII,* and *D-VIII* SNPs, not in the minor alleles. High retinol intake appeared to reduce obesity risk in the minor alleles of the *D-I*, *D-VII*, and *D-VIII* SNPs, even though intake of animal fats and retinol remained higher among minors over the three years. Conclusions: These findings suggest that the RMR/BW ratio and dietary fat/retinol intake should be considered in *DNACJ6*-gene-based precision medicine approaches for pediatric obesity prevention, particularly for boys.

## 1. Introduction

Resting metabolic rate (RMR), which accounts for 60–70% of total daily energy expenditure, refers to the calories burned when your body is completely at rest [1]. RMR supports basic physiological functions such as breathing, blood circulation, organ functions, and neurological functions. RMR can be measured using direct or indirect calorimetry, but the most common methods utilized today are mathematical formulars such as the amended Harris and Benedict (H&B) equation (1984) and the Mifflin-St Jeor formular created in 1990 [2,3,4]. Measuring RMR is crucial because it is often used interchangeably with basal metabolic rate (BMR), though BMR, typically 10–20% lower than RMR, is essential in establishing strategies for obesity prevention programs [5]. Among the many factors affecting RMR, such as body composition, body surface area, body temperature, age, sex, genetics, pregnancy, hormonal status, and energy restriction, the most important determinant is lean body mass (LBM) or fat-free mass (FFM) [6]. However, it remains unclear whether those factors play a role in early-onset obesity due to an imbalance of energy homeostasis.

To investigate the relationship between genes associated with RMR and pediatric obesity, we previously conducted three studies. Firstly, we screened SNPs in RMR-associated genes, such as *mitogen-activated protein kinase 6 (MAP2K6* or *MEK6)* and *DnaJ heat shock protein family (Hsp40) member C6 (DNAJC6)*, using genome-wide associated study (GWAS) to assess their association with child obesity risk [7]. In a subsequent study, we found that *DNAJC6*-overexpressed 3T3-L1 cells (Tg*^Hsp^*) depressed expression of adipogenesis-related genes, such as PPARγ, C/EBPα, and aP2 (adipogenesis-related biomarkers), as well as reducing adipokines expression and IRS signaling [8]. In DNAJC6 knockdown experiments, we observed fat synthesis during differentiation (PPARr, C/EBPa, aP2, etc.) and heightened intracellular stress, which was linked to reduce mitochondrial respiratory capacity [9]. In the case of *MAP2K6*, our Tg*^MEK6^* model with *MEK6* overexpression revealed increased fat accumulation and inflammation due to lipolysis inhibition [10]. In a 3-year panel study, we identified that high-risk factors of child obesity, such as SBP, WC, and dietary fat intake, should be controlled, particularly in boys with mutants of *MEK6* SNPs, *rs9916229*, or *rs756942,* which may help prevent pediatric obesity [7]. These studies led us to hypothesize that RMR-based energy imbalance may be present in human obesity with *DNAJC6* SNP mutations.

Mutations in *DNAJC6* gene are associated with mental retardation and early-onset Parkinson disease [11,12,13]. Vauthier et al. found that 7-year-old children with a homozygous 80 kb deletion in the chromosomal 1p31.3 region were associated with early onset obesity, mental retardation, and epilepsy [14]. The deleted region comprises the proximal promoter and exons 1 and 2 of the LEPR gene and exons 5 to 19 of the *DNAJC6* gene. This finding, where seven of eight heterozygotes carrying the 80 kb deletion were overweight, is consistent with previous observations that human carriers of mutation in the leptin or leptin receptor genes are predisposed to overweight and obesity [15]. Based on the variation in *DNAJC6* on chromosome 1 (65460411-65495105 region), which is related to early-onset obesity, we designed a three-year panel study to investigate whether *DNAJC6* variation related to RMR alteration affects risk factors of obese environments in children aged 8–9 years old.

## 2. Methods

### 2.1. Study Population

A total of 807 subjects aged 8–9 years were recruited from eight elementary schools located in Guro-ku in Seoul, Korea, during a regular health check-up period. The 3rd grade students from these schools in 2009 (baseline) were followed up (F/U) at 12 years old, when they were in the 6th grade, in 2012 (3 years later). During the F/U period, 295 subjects were excluded due to medical issues, being already overweight and obese (OB) at baseline, parental refusal to participate in the F/U process, or missing at least one examination, including incomplete, undetectable, or unreliable records. The final study included 512 subjects (boys = 372 and girls = 140; 63.3% F/U rate), who were enrolled after exclusions. These participants underwent a basic survey that included questions about their parent’s health history, body mass index (BMI), education and income, physical activity, and dietary intake. Data were collected for each child at ages 9 and 12. At the end of the study, we classified children as OB, including overweight, and non-OB according to BMI criteria. Obesity was defined as children having a BMI of the 85th percentile and above using the ‘Korean National Growth Charts’. After genotyping of 9 DNAJC6 in all students, we investigated how RMR affected the change in variables, ((2012−2009)/2012), such as anthropometric measurements, blood biochemistry, and dietary habits in OB and non-OB groups.

Institutional review board (IRB) approval was granted by Korea University, Guro Hospital (No.GR0837-001), for the study protocol and Sungshin Women’s University (No. SSWUIRB 2012-015) for the new SNPs analysis in 2012. Written informed consent was obtained from both the children and their parents. The flowchart of the 3-year panel study accounting the subjects is shown in Figure 1.

### 2.2. DNAJC6 Selection as RMR-Related Gene (GWAS) and Genotyping

Using Affymetrix Genome-Wide Human SNP array 6.0 in the DNA Link (Sedamon-ku, Seoul, Republic of Korea), we analyzed which gene was associated with BMR and BMI in the small children group (*n*= 113). We performed systematic quality control steps on the raw genotype data and obtained 904,085 SNPs. SNPs with a minor allele frequency of <1%, a call rate of <95%, and a significant deviation from Hardy–Weinberg equilibrium (HWE) in controls (*p* < 1 × 10^−7^) were excluded. After quality control, 700,000 SNPs were included in this study, and a total of 21 genes and 54 SNPs were found to be highly associated with RMR and BMI, reaching formal genome-wide statistical significance with the threshold *p*-value. A few genes were commonly associated with both RMR and BMI, and *DNAJC6* was selected as a primary subject for research due to a homozygous deletion comprising part of *DNAJC6* and *LEPR* genes on chromosome 1 (region 65460411-65495105), which is associated with early-onset obesity [14]. Nine SNPs of *DNAJC6* gene were analyzed by SNaPshot methods (ABI PRISM SNaPShot Multiplex kit, Foster City, CA, USA) and assigned serial numbers instead of rs-number for convenience, such as *D*-*I (rs17127601)*, *D*-*II (rs6697554)*, *D*-*III (rs1359530)*, *D*-*IV (rs3850826)*, *D*-*V (rs6588132)*, *D*-*VI (rs10789182)*, *D*-*VII (rs1334880)*, *D*-*VIII (rs7354899)*, and *D*-*IX (rs1334881)*. The genomic DNA flanking the SNPs was amplified using PCR reaction with forward and reverse primer pairs (Appendix A). Analysis was carried out using GeneMapper SW (version 4.0; Applied Biosystems, Foster City, CA, USA).

### 2.3. Collection of Anthropometric, Biochemistry, and Dietary Intakes Data

Standing height (Ht), body weight (BW), waist circumference (WC), and systolic and diastolic blood pressure (SBP and DBP; NISSEI, Hanishina-gun, Nagano-ken, Japan) were measured. BMI was calculated by dividing BW (kg) by the square of Ht (m^2^). We classified obesity (OB), including overweight, and non-OB in children by obesity criteria, evaluated 3 years later. The predictive RMR of H&B equation (boys, 66.5 + 13.8Wt + 5Ht − 6.8A; girls, 655.1 + 9.6Wt + 1.9Ht − 4.7A, as A stands for age) was used in this cohort study. In a previous study, we found the H&B method for RMR calculation was close to 60% of total energy requirement following IOM equation for Korean elementary students [7]. Whole blood samples were obtained from genotyping and serum used for blood biochemistry using a Hitachi-7600 analyzer (Hitachi, Tokyo, Japan), such as total cholesterol (TC), triglyceride (TG), high-density lipoprotein cholesterol (HDLc), and fasting blood glucose (FBS). Low-density lipoprotein cholesterol (LDLc) levels were calculated using the equation LDLc = TC – HDLc − (TG/5). Commercial ELISA Kits (Mesdia, Seoul, Republic of Korea) were used for estimating fasting insulin levels. For dietary data collection, trained dietitians explained the method used to complete the 24 h dietary recall. Dietary intakes for 3 days (2 weekdays and 1 weekend day) were recorded by a trained interviewer and the food records were crosschecked with the parents or guardians of children. CAN-Pro4.0 (Korean Nutritional Society, Seoul, Republic of Korea) was used for the quantitative analysis of nutrients based on food records.

### 2.4. Statistics

For the analysis of GWAS data, the subjects were divided into two groups: a control (non-OB) group with high RMR and low BMI and a case (OB) group with low RMR and high BMI. Continuous variables were expressed as mean ± SD, and differences between groups were assessed using Student’s *t*-test. Categorical variables were represented as percentages and tested using the χ^2^ test. Hardy–Weinberg equilibrium (HWE) was assessed using the χ^2^ test, with a *p*-value > 0.05 considered significant. All datasets were filtered to exclude samples or SNPs with >5% missing values, variants with <5% minor allele frequencies (MAFs), and samples deviating from the HWE, using PLINK, a whole-genome data analysis toolkit. Quantitative trait association analyses for RMR and BMI were performed in the OB (RMR ≒ 1023 kcal/day and BMI > 85th percentile) and non-OB subjects (RMR ≒ 1043.8 kcal/day and BMI < 85th percentile). BMI and RMR were analyzed as continuous traits separately in OB and non-OB subjects with linear regression in PLINK, using dominant, codominant, and recessive models. 

Variable differences between the baseline and the three-year follow-up were tested by the paired *t*-test, while differences between non-OB and OB were assessed with Student *t*-test. The SNP-related variable differences were performed with an independent-sample *t*-test under the assumption of equal variance. Welch’s *t*-test was applied when the equal variance assumption was not met. Changes in obesity prevalence according to RMR or RMR/BW were assessed using the Cochran Armitage trend test or the Jonckheere–Terpstra test as appropriate. Stepwise multiple regression analysis was used to identify the risk factors that increase BMI in relation to nine *DNAJC6* SNPs. Data analysis was conducted using SAS software version 9.1.3 (SAS Inc., Cary, NC, USA) and PLINK [16]. Additionally, odds ratios (OR) were calculated to assess the strength of the association between the risk of obesity (including overweight) and the variable changes according to the nine *DNAJC6* SNPs.

## 3. Results

### 3.1. Differences in the Variables Between Non-OB and OB Groups for 3 Years

During the three-year study, increases were observed in various anthropometrics, blood chemistry, and dietary intake variables. The rate of overweight/obesity in 2012 (18.4%) decreased by 3.2% compared to 2009 (21.8%), although the mean BMI increased in 2012 (19.5) compared to baseline (18.2). BMI, WC, TG, and HDL increased, while plasma LDL, FBS, AST, and ALT significantly decreased among total subjects (Table 1). However, the levels of blood parameters in total subjects were not concerning enough to cause metabolic problems. Dietary intakes of total calories, proteins (plants and animal), lipids (plants and animals), vitamins (B_1_ and B_6_), and minerals (P, Fe, and Zn) increased. Conversely, polyunsaturated fatty acids (PUFA), vit C, and calcium intake were significantly decreased. However, the percentages of energy for carbohydrate (57–58%), protein (15%), and fat (27%) were not changed before and after 3 years. Cholesterol, saturated fatty acids, monounsaturated fatty acids (MUFA), vitamins (B2, folate, niacin, A, and E), and minerals (Na and K) showed no change. All dietary intakes were adjusted according to total energy intake in 2009 and 2012, respectively.

After OB and non-OB groups were classified at the study endpoint, the differences over 3 years were analyzed by paired *t*-test with comparison between non-OB and OB using Student *t*-test (Table 1). In the OB group, BMI, WC, and SBP were significantly higher, while plasma HDLc was lower compared to non-OB children for 3 years. OB subjects had significantly lower intakes of total calories, carbohydrates, vitamins (B_2_, C, folate, A, and retinol), and minerals (Fe and Zn) than non-OB children. Since no formal intervention or nutritional education was conducted over 3 years, it is possible that the OB children tried to avoid becoming more obese.

Average RMR increased by 260.3 kcal for 3 years, with greater increases in OB group (347.7 kcal) compared to non-OB group (172.8 kcal), likely due to the growth period. In a previous study, RMR increased significantly with BMI over 3 years, showing a positive correlation between BMI and RMR (BMI = 15.2 + 0.0042 RMR, R2 = 30.0%, *p* = 0.0013) [17]. In this study, the positive correlation between RMR and BMI was stronger in non-OB subjects compared to OB subjects (Figure 2A). Interestingly, a negative correlation between RMR/BW and BMI in non-OB group (R^2^ = 59.1%) was stronger compared to the OB group (R^2^ = 37.82%), contrasting the positive correlation between RMR and BMI (Figure 2B). Although a negative correlation between RMR and BMI is often expected in adults, this pattern did not apply in children due to their active growth period. Therefore, RMR/BW may be a more suitable variable than RMR in studies on childhood obesity.

### 3.2. RFs of Obesity with Gender Differences According to DNAJC6 SNPs

The MAFs of nine *DNAJC6* SNPs coded as (major/minor) were as follows: *D-I (A/G)*, *D-II (A/G)*, *D-III (A/G)*, *D-IV (C/G)*, *D-V (C/T)*, *D-VI (T/C)*, *D-VII (G/A)*, *D-VIII (C/T)*, and *D-IX (G/C)*, with MAFs of 8.4%, 19.6%, 19.4%, 19.5%, 19.2%, 19.2%, 19%, 19.2%, and 19.3%, respectively. In the case of two SNPs (*D-VI* and *D-IX*), major alleles were mutant types, but the major alleles of the other seven SNPs were the wild types. Since the range of genotype frequencies for which the HWE is satisfied, statistical analysis for minor alleles’ effects on overweight/obese prevalence was valid. 

To analyze the association with obesogenic environments in obesity according to RMR-related *DNAJC6* SNPs, we need to know what allele types of nine SNPs were involved in the prevalence of children obesity at the end of this cohort. The results of the χ^2^-test showed significant differences in the relative frequencies (RFs) of obesity between major and minor allele types in *D-I*, *D-VI*, *D-VII*, *V-III*, and *D-IX* SNPs in total subjects (Table 2). Notably, obesity RF for *D-I* was 40%, nearly double that of the minor alleles of the other SNPs, despite *D-I* having a lower MAF of 8.4% compared to an average of 19% for the other SNPs. This pattern was especially pronounced in boys, with 43.3% RFs of obesity among those carrying *D-I*, while no significant differences were observed in girls.

### 3.3. The Differences in Changes in Obesity Variables Between Major and Minors of Nine SNPs

The averages of RMR were significantly increased by 260.3 kcal over 3 years due to the growth period. However, we found changes in RMR, BMI, BW, and RMR/BW were significantly higher in children with a minor allele of only *D-I* SNP, compared to those with the major allele (Figure 3A). The RMR, BW, or BMI in the other eight SNPs did not change by alleles. 

As observed in the results of RFs for obesity, the changes in RMRs in total subjects were similar in boys but not in girls. The changes in BMI for 3 years were higher in minor allele (2.25) than in those with the major allele (1.75) of D-I SNP in both total subjects and boys but not in girls. We concluded RMR, which accounts for 60–70% of total energy expenditure, was increased in children with the minor mutant allele of *D-I* SNP, which may be associated with a higher prevalence of obesity. Anthropometric variables such as RMR, BW, and RMR/BW were considered high risk factors in the minor mutant allele of *D-I* SNP. On the other hand, diet variables related to dietary fats were found to be risk factors in minor alleles of *D-VII* and *D-VIII* SNPs. In the case of the *D-VII* and *D-VIII* SNPs, the changes in animal fat, SFA, and retinol intakes were 3.5 to 4 times higher in individuals with the minor alleles compared to those with the major alleles (Figure 3B).

### 3.4. The Significant Risk Variables to Increase Child Obesity with Mutant Alleles of Three SNPs

Using logistic regression with the above factors to assess the risk of obesity, we aimed to identify the variables that contribute to an increased risk of obesity. We found the minor alleles of three SNPs, such as *D*-*I*, *D*-*VII*, and *D*-*VIII*, and RMR were high risk factors to increase child obesity among the nine SNPs. In contrast, neither the minor allele nor RMR was found to be a risk factor for increasing BMI in the other six SNPs (Table 3). In children with a mutant allele of *D*-*I* SNP, OR for the BMI risk was significantly increased by 2.59 times (CI; 1.0680~6.2740) compared to those with a wild allele. The OR for BW increase in obesity in children with mutant *D*-*I* SNP was 1.57 (CI; 1.4110~1.7560) times higher than that in the wild type. The ORs in mutant alleles of *D*-*VII* and *D*-*VIII* had 1.86 (CI; 1.010~13.422) and 1.85 (CI; 1.008~3.416) times higher obesity risk than in wild allele compared to their major alleles. However, the ORs for obese risk were significantly reduced by almost 50% in mutants with both SNPs, such as *D*-*VI* (0.54, CI; 0.293~0.991) and *D*-*IX* (0.55, CI; 0.298~0.992), compared to wild types of these. Since the mutant types of both D-VI and D-IX SNPs were the major alleles, the opposite results of *D*-*I*, *D*-*VII*, and *D*-*VIII* SNPs might appear.

Although diet variables did not show significant differences between the alleles of *D-I* SNP, it is advisable to recommend reducing animal fats, including trans fatty acids (TFA) and saturated fatty acids (SFA), to control BMI in children with the minor allele of *D*-*I*. Although RMR and dietary retinol intake were similar between mutant and wild types, high intake of retinol may reduce the risk of obesity in all three *D*-*I*, *D*-*VII*, and *D*-*VIII* SNPs (0.99, CI; 0.990~1.000). The other variables, such as animal fat intake, TFA, and SFA, which differed between wild and mutant types for 3 years, were not found to increase the risk of obesity.

Using the Cochran–Armitage test, the prevalence of obesity was significantly decreased by increasing the changes in RMR/BW for 3 years in major alleles of three SNPs, *D*-*I*, *D*-*VII*, and *D*-*VIII*, compared to minor alleles (non-significant). When the changes in RMR/BW were divided by quartiles (Q1–Q4; <4.29, 4.3–5.2, 5.21–7.19, and >7.2), the risk of obesity was significantly decreased (*p*_-trend_ < 0.05) with an increase in the RMR/BW in major alleles (Figure 4). The OR for risk of obesity in Q4 was reduced to 0.34 (CI; 0.16–0.72) compared to Q1. While the high risk of obesity was not statistically significant in the lower values of RMR/BW in the mutant allele of *D-I* SNP, the correlation between RMR and BMI was highest in the mutant allele of *D*-*I* compared to the other eight SNPs. RMR and BW were high-risk variables for increasing obesity in children with the mutant type of *D*-*I* SNP. Therefore, RMR/BW and retinol intake might be used as biomarkers for the RMR-related *DNAJ6*-gene-based precision nutrition to prevent child obesity in further intervention.

## 4. Discussion

The prevalence of obesity increased twice (8.7% to 15.0%) for 10 years (2007 to 2017) in children aged 6–18 years [17]. According to the results of regular medical check-up in 2019, one of four students (25.8%) in elementary, middle, and high schools (n = 104,380, 1023 schools sampled) could be overweight or obese. The environmental risk factors of childhood obesity, such as diet, physical activity, life habits, and genetics, need to be identified, as childhood obesity is associated with early-onset adult chronic diseases, leading to increased morbidity and mortality [18].

In a previous study, we discussed that researchers are focusing on finding predictive RMR methods that are suitable for their own population, especially children. We found the Harris and Benedict method for RMR calculation was close to 60% of total energy requirement following IOM equation for Korean elementary students [7]. Using significant RMR genes related to obesity founded by GWAS, such as *MAP2K6* and *DNAJC6*, we identified various factors affecting RMR-based child obesity. We found BW and dietary fat/retinol intakes should be considered to prevent child obesity (BMI), especially in children with minor alleles of *DNAJC6* SNPs, such as *rs17127601 (D-I)*, *rs1334880 (D-VI)*, or *(rs7354899 (D-VIII)*, particularly in boys. The changes in BMI, RMR, WC, and BW were higher in OB compared to non-OB, and RMR was proportional to BMI and BW. The risk of BMI was increased by decreasing RMR/BW, consistent with previous studies [7]. Bernstein et al. reported that, due to higher body fat (33.6%) in obese subjects, RMR was higher in obesity than non-obese individuals [19]. The increase in RMR in obesity is an important mechanism for achieving energy balance, while the decrease in RMR during wight loss demonstrates the need for reduced food intake to maintain energy balance once normal BW is reached [20]. Although we did not have body composition data, we hypothesized that obese subjects with higher RMR may have an overall better metabolic profile compared to obese individuals with low RMR [21]. We found the changes in BMI, RMR, and BW were higher in children with minor (mutant) alleles than major (wild) in *D-1 (rs17127601).* In META study, it was suggested that, whether low RMR in relation to body size and composition is genetic or acquired, the existence of a low RMR is likely to contribute to a high rate of BW regain [22]. Measuring only RMR in the obese state is unlikely to provide a full understanding of the pathogenesis of obesity; therefore, other indicators, such as RMR/BW, might be better. Adjusted RMR/BW or, to a lesser extent, BMI allows for better comparison between individuals of various body sizes, as BW or BMI contribute to RMR in a consistent fashion over a wide range of body sizes or BMI values [23]. RMR/BW index was significantly correlated with higher-energy-density diets, such as high intake of fat, lean meats, and tea/coffee and low intake of vegetables and fruits, although age, FFM or LBM, and physical activity were adjusted [24]. In our study, the increase in RMR associated with consumption of high-energy-dense diets may be due to the body’s effort to maintain sustainability. With a strong relationship between RMR or RMR/BW and energy intake, foods containing animal fat, TFA, or SFA may increase RQ, VO2, VCO2, and, consequently, elevate RMR or RMR/BW.

The *DNAJC6* gene encodes auxilin on chromosome 1, a neuronal protein that functions specifically in the pathway of clathrin-mediated endocytosis [25]. *DNAJC6* gene belongs to the evolutionarily conserved DNAJ/HSP40 family of proteins, which regulate molecular chaperone activity by stimulating ATPase activity [26]. *DNAJC6* gene is abundantly expressed in various human brain regions, but there was very low expression in non-neural tissues, such as liver, adipose tissue, muscle, and bone marrow [12]. Mutations in the DNAJC6 gene have been associated with mental retardation and Parkinson disease; early-onset obesity, mental retardation, and epilepsy were found in 7-year-old children with a homozygous 80 kb deletion in the chromosomal 1p31.3 region [12,13,14]. The deleted region comprises the proximal promoter and exons 1 and 2 of the LEPR gene and exons 5 to 19 of the *DNAJC6* gene. Interestingly, seven of eight heterozygotes carrying the 80 kb deletion were overweight, which is consistent with previous observations that heterozygous human carriers of leptin or leptin receptor mutations are predisposed to overweight and obesity [15].

Previous research has shown that SNPs in leptin-receptor-related genes like *DNAJC6* are strongly associated with obesity-linked cancers, including pancreatic cancer and postmenopausal breast cancer, although circulating leptin was not related with cancer risk [27,28]. Leptin is involved in regulating food intake, energy expenditure, and glucose and fat metabolism as a potential biological link in the development of obesity. This could explain the association between BMI and RMR. Recent studies have demonstrated that individuals with low RMR, who are theoretically at greater risk for obesity-related disorders, have higher serum adiponectin levels, which is the strongest predictor of RMR, followed by insulin resistance [29]. Moradi et al. reported that circulating adipokines like vaspin are higher in obese individuals and mediate the association between visceral fat and FM with RMR in obese participants [30]. In our previous in vitro study, we found adipogenesis genes were depressed in *DNAJC6*-overexpressed 3T3-L1 cells (Tg*^Hsp^*), as well as the expression of the adipokines leptin and adiponectin released by adipose tissues. We also found insulin receptor expression and IRS signals, such as AKT phosphorylation and GLUT4 in Tg*^Hsp^* cells [8]. Drabsch et al. reported RMR was highly associated with homeostasis model assessment of insulin resistance (HOMA-OR) after adjustment for FFM, sex, age, and education, but inflammatory markers like C-reactive protein did not mediate this association in a large cohort [31]. Since we found the intracellular stress level was increased by mitochondria dysfunction in the inhibition of *DNAJC6* gene, the production of proinflammatory cytokines may change with RMR fluctuation [9]. Given that inflammation and oxygen uptake (VO2max) are significantly associated with major diseases, one modifiable factor contributing to inflammation is diet. In the dietary inflammatory index (DII) study, RMR/FFM in obesity was inversely associated with DII score [32]. High-DII (proinflammatory foods) included energy intake, carbohydrate, cholesterol, total fat, FA (trans and saturated), protein, vit B-12, and Fe, but the low-DII (anti-inflammatory foods) included dietary fiber, PUFA (n-6, n-3), vitamins (A, D, C, E, B complex, and folate), minerals (Zn, Mn, and Se), etc. (onion, tea, and garlic). Vit A signaling can regulate pathways adversely affected in states of obesity, such as hepatic lipid metabolism, pancreatic endocrine functions, adipogenesis, and immune functions [33]. Korean DRI recommended 550–600 ug RAE for 9–11-year-old children, but OB children have been recommended to have a lower intake of Vit A compared to non-OB. Retinol intake has been shown to be a significant variable in suppressing obesity in children with the *DNAJC6* mutant; the chemical structure of vitamin A may also be relevant. Dietary Vit A depletion or genetic manipulation of Vit A signaling can promote adipose tissue expansion, suggesting that dietary or pharmacological administration of Vit A or its metabolites can reduce obesity and promote energy utilization in obesity. While blood retinol levels have not yet been established as a universal risk biomarker for obesity, plasma retinol is a recognized risk factor for obesity in Brazilian children [34]. On the other hand, serum retinol-binding protein-4 is positively associated with CVD, T2DM, and obesity, potentially contributing to the development of obesity-related co-morbidities [35]. Given that RMR is increased in the mutants of both *MEK6 and DNAJC6* gene, which were high risk variables of child obesity, proinflammatory foods, such as those high in total energy intake and fat, should be excluded. Instead, anti-inflammatory food, such as vit A (retinol), should be recommended in the meals to help prevent child obesity. 

The strength of our study lies in being the first to report how the variation in RMR-related gene *DNAJC6* modulates overweight/obesity in Korean children. However, several limitations should be considered, particularly in the context of a prospective study. Firstly, we used anthropometric measurement for body composition in children rather than more precise methods, such as DEXA and bioelectrical impedance (BIA). DEXA, the standard method, was not available for ethical reasons in children of 8~12 years. Additionally, we lacked access to a BIA device calibrated with child-specific reference for 8-year old in 2009. Given the principle of consistency in research methodology, it was difficult to apply modified BIA, even after 3 years. Secondly, we used the predictive method for estimating RMR instead of the indirect calorimetry, as many parents were opposed to prolonged testing on children. Thirdly, due to the small sample size of the OB group, we were unable to assess which test method (high vs. low RMR) would have been more metabolically relevant for obese children. Finally, adjusted methods (repeated measures correlation) were not performed to assess the within-individual association to avoid type-I error. 

## 5. Conclusions

We suggest that *DNAJC6*, along with *MEK6*, as RMR-related genes, provides new insights into energy balance. Increased changes in RMR, BMI, BW, and RMR/BW were observed with mutant allele of *D-I* SNP, which was also associated with a higher prevalence of obesity. Greater increases in animal fat intake, including saturated fatty acids and retinol, were noted in the minor alleles of *D-VI*, *D-VII*, *D-VIII*, and *D-IX* SNPs compared to the major alleles. These findings suggest that the RMR/BW ratio and dietary fat, including retinol intake, should be considered in *DNACJ6*-gene-based precision nutrition aimed at preventing child obesity in relation to energy imbalance associated with RMR, particularly for boys.

## Figures and Tables

**Figure 1 nutrients-16-04423-f001:**
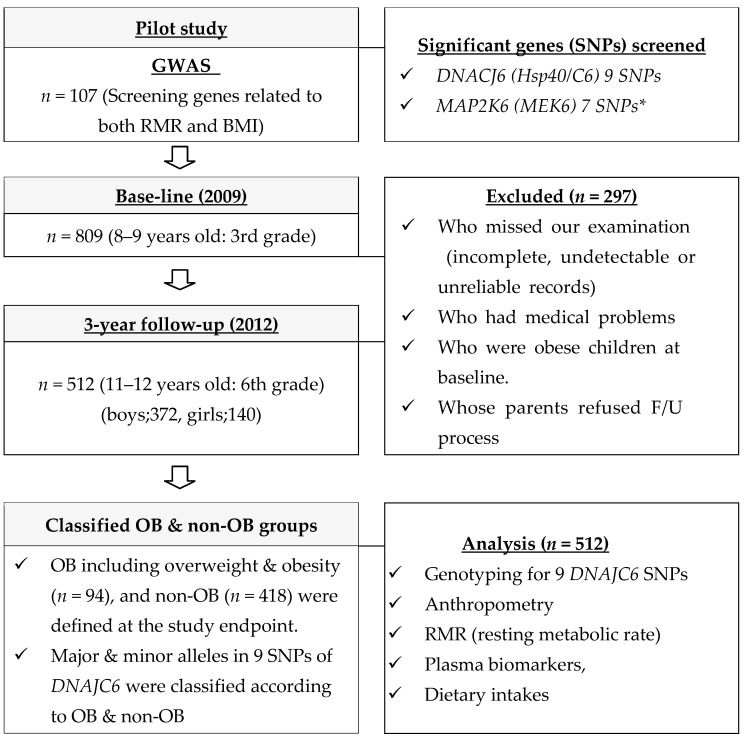
Flowchart of the 3-year panel study accounting the subjects. (*; [7]).

**Figure 2 nutrients-16-04423-f002:**
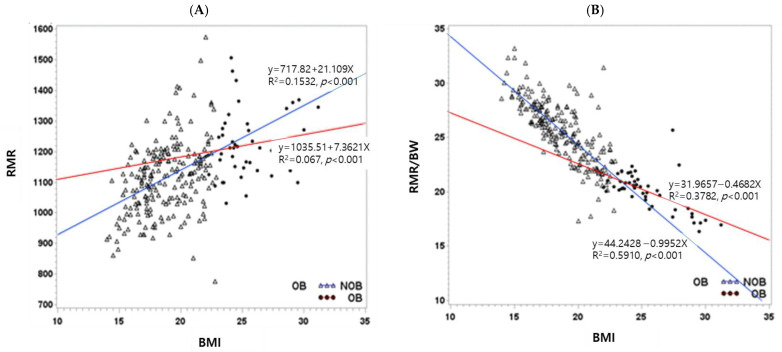
The positive correlation between RMR and BMI (**A**) and the negative correlation between RMR/BW and BMI (**B**). The negative correlation between RMR/BW and BMI was more powerful than the positive correlation between RMR and BMI. The regression model for non-OB subjects (blue line) was significantly more explanatory than that of OB subjects (red line) in both A and B cases. The negative correlation between RMR and BMI in non-OB group (R^2^ = 59.1%) was more powerful compared to OB group (R^2^ = 37.82%).

**Figure 3 nutrients-16-04423-f003:**
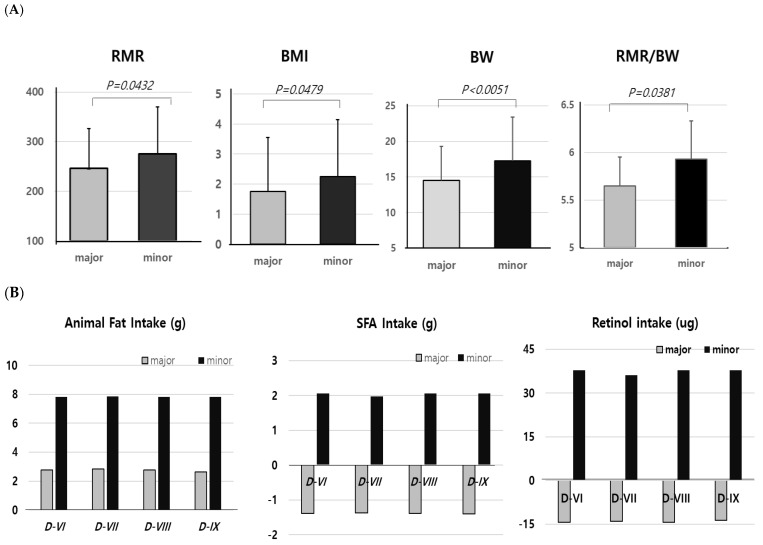
The significant differences in changes in variables in major (wild) and minor (mutant) alleles of *D-1* (*rs17127601)*. (**A**) The changes in RMR, BMI, or BW were higher in minor than major allele of only *rs17127601* SNP (Welch’s T-test), but the other minor alleles of 8 SNPs did not change RMR, BMI, and BW compared to majors. Significant differences in the changes in dietaty intakes between major and minor alleles of *D*-*VI, D*-*VII, D*-*VIII*, and *D*-*IX* SNPs. (**B**) Diet variables related to fat intake were risk variables in minor allele of *D*-*VI (rs10789182), D*-*VII (rs1334880), D*-*VIII (rs7354899)*, and *D*-*IX (rs1334881)* SNPs. All dietary intakes were adjusted according to energy intake in 2009 and 2012, respectively. The ranges of SD values in major and minor were (17.89~18.07 and 22.81~22.92) for animal fat intake, (8.63~8.67 and 13.61~13.65) for SFA intake, and (153.35~153.73 and 196.80~197.15) for retinol intake.

**Figure 4 nutrients-16-04423-f004:**
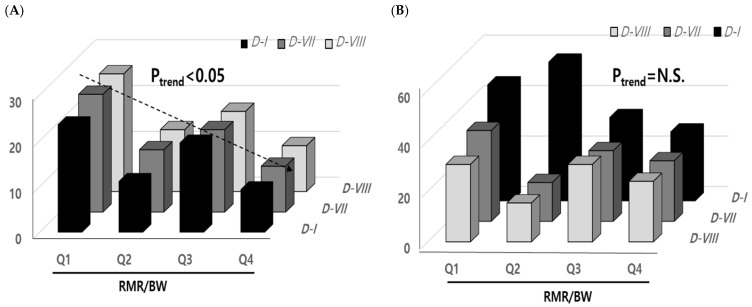
Relative frequency of obesity (%) in the quartiles of RMR/BW changes of majors (**A**) and minors (**B**) in *D-1, D-VII*, and *D-VIII* SNPs. By the Cochran–Armitage test, the prevalence of obesity was significantly decreased (*p*-^trend^ < 0.05) by increasing the changes in RMR/BW for 3 years in major alleles only in all three SNPs, D-I, VII, and D-VIII, compared to minor alleles. In major alleles, the averages of quartile values (Q1~Q4) of RMR/BW changes for 3 years were Q1 (<4.3), Q2 (~5.2), Q3 (~7.19), and Q4 (>7.2) compared to the minor alleles (Q1, <4.3; Q2, ~5.3; Q3, ~7.49; and Q4, >7.5) (*D-I (rs17127601), D-VII(rs1334880),* and *D-VIII (rs7354899)*).

**Table 1 nutrients-16-04423-t001:** Differences in follow-up variables between OB and non-OB subjects for 3 years.

Variables (Units)		Total ^(1)^			Non-OB			OB			*p*-Value ^(2)^ (non-OB vs. OB)
		n	Means	SD	n	Means	SD	n	Means	SD	
Anthropometrics	BMI (kg/m^2^)	512	1.79 †	1.78	418	1.49	1.48	94	3.17	2.30	***
	RMR	512	260.3	32.60	418	172.8	30.3	94	347.7	34.8	***
	WC (cm)	512	9.55 †	6.42	404	8.73	5.60	94	13.06	8.32	***
	SBP (mmHg)	512	0.36	19.71	416	−0.69	18.76	94	5.00	23.00	*
	DBP (mmHg)	512	−2.66 †	12.23	416	−3.28	12.38	94	0.09	11.18	NS
Blood chemistry	TC (mg/dL)	418	0.50	27.47	338	−0.40	26.08	80	4.30	32.61	NS
	LDL (mg/dL)	418	−10.35 †	28.39	338	−11.75	26.49	80	−4.43	34.92	NS
	TG (mg/dL)	418	5.82 †	53.12	338	6.40	50.50	80	3.38	63.34	NS
	HDL (mg/dL)	418	9.68 †	9.19	338	10.07	9.45	80	8.05	7.82	*
	FBS (mg/dL)	418	−4.84 †	10.72	338	−4.59	10.70	80	−5.90	10.80	NS
	AST (U/L)	418	−5.73 †	9.00	338	−5.28	5.85	80	−7.63	16.65	NS
	ALT (U/L)	418	−4. 28 †	17.40	338	−3.20	7.75	80	−8.85	36.27	NS
Dietary Intake ^(3)^	Total Calories (kcal)	488	261.88 †	599.26	402	291.34	610.36	86	124.18	526.07	*
	Carbohydrate (g)	488	40.72 †	95.73	402	45.72	97.24	86	17.34	84.99	*
	Protein (g)	488	10.95 †	50.93	402	11.58	54.39	86	7.98	29.94	NS
	Fat (g)	488	5.99	27.11	402	6.73	27.97	86	2.54	22.48	NS
	Cholesterol (mg)	488	−19.90	250.06	402	−14.65	257.52	86	−44.46	211.34	NS
	Fiber (g)	488	0.14	6.52	402	0.36	6.55	86	−0.90	6.35	NS
	TFA (g)	488	−2.07	24.02	402	−1.73	24.84	86	−3.68	19.78	NS
	SFA (g)	488	−0.69	9.93	402	−0.50	10.29	86	−1.59	8.10	NS
	MUFA (g)	488	−0.42	9.57	402	−0.37	9.94	86	−0.68	7.58	NS
	PUFA (g)	488	−0.95 †	5.55	402	−0.86	5.67	86	−1.41	4.95	NS
	Vit B1 (mg)	488	0.23 †	0.59	402	0.24	0.61	86	0.15	0.46	NS
	Vit B2 (mg)	488	0.06 †	0.59	402	0.09	0.59	86	−0.07	0.56	*
	Vit B6 (mg)	488	0.26 †	0.86	402	0.28	0.84	86	0.15	0.93	NS
	Niacin (mg)	488	3.56 †	7.30	402	3.77	7.51	86	2.58	6.09	NS
	Vit C (mg)	488	−14.12 †	55.13	402	−10.12	53.01	86	−32.80	61.09	***
	Folate (µg)	488	3.45	120.11	402	11.17	120.82	86	−32.62	110.46	***
	Vit E (mg)	488	−0.11	7.705	402	0.20	7.64	86	−1.57	7.87	NS
	Vit A (µgRAE)	488	−9.95	504.61	402	12.86	509.05	86	−116.55	471.62	*
	Retinol (µg)	488	−3.86	164.41	402	3.23	123.69	86	−32.88	96.88	*
	Ca (mg)	488	−46.51 †	333.28	402	−40.36	326.27	86	−75.25	364.92	NS
	phosphate (mg)	488	61.27	429.44	402	71.61	430.94	86	12.92	421.50	NS
	Fe (mg)	488	1.46	8.18	402	2.07	8.47	86	−1.40	5.91	***
	Na (mg)	488	−133.32	1554.85	402	−111.25	1582.56	86	−236.50	1422.26	NS
	K (mg)	488	−10.79	1108.41	402	26.27	1089.83	86	−184.01	1182.76	NS
	Na/K	488	−0.01	0.61	402	−0.03	0.62	86	0.09	0.57	NS
	Zn (mg)	488	0.99	3.48	402	1.22	3.51	86	−0.05	3.13	**

^(1)^ †; Significant differences in variables for 3 years in total subjects with *p*-value < 0.05. ^(2)^
*p*-values for comparison of non-OB and OB; *: <0.05, **: <0.01, ***: <0.001. NS; non-significance (pared *t*-test). ^(3)^ All dietary intakes were adjusted according to energy intake in 2009 and 2012, respectively.

**Table 2 nutrients-16-04423-t002:** The relative frequencies (%) of obesity with gender differences according to nine DNAJC6 SNPs in non-Ob and OB groups categorized at the end point.

SNP	rs Number	Genotype ^(1)^ (Major vs. Minor)	MAF ^(2)^	Total						Boys						Girls					
				Non-OB (%)		OB (%)		Total	^(3)^ *p*- Value	Non-OB (%)		OB (%)		Total	^(3)^ *p*- Value	Non-OB (%)		OB (%)		Total	^(3)^ *p*- Value
*D*-*I*	*rs17127601*	A		389	84.0	74	16.0	463	***	293	85.7	49	14.3	342	***	96	79.3	25	20.7	121	NS
		G	8.4	27	60.0	18	40.0	45		17	56.7	13	43.3	30		10	66.7	5	33.3	15	
*D*-*II*	*rs6697554*	A		338	83.0	69	17.0	407	NS	252	84.6	46	15.4	298	NS	86	78.9	23	21.1	109	NS
		G	19.6	78	75.7	25	24.3	103		56	77.8	16	22.2	72		22	71.0	9	29.0	31	
*D*-*III*	*rs1359530*	A		336	83.0	69	17.0	405	NS	252	84.6	46	15.4	298	NS	84	78.5	23	21.5	107	NS
		G	19.4	76	75.2	25	24.8	101		54	77.1	16	22.9	70		22	71.0	9	29.0	31	
*D*-*IV*	*rs3850826*	C		337	83.0	69	17.0	406	NS	251	84.5	46	15.5	297	NS	86	78.9	23	21.1	109	NS
		G	19.5	77	75.5	25	24.5	102		55	77.5	16	22.5	71		22	71.0	9	29.0	31	
*D*-*V*	*rs6588132*	C		340	83.1	69	16.9	409	NS	254	84.9	45	15.1	299	NS	86	78.2	24	21.8	110	NS
		T	19.2	78	75.7	25	24.3	103		56	76.7	17	23.3	73		22	73.3	8	26.7	30	
*D*-*VI*	*rs10789182*	C		341	83.4	68	16.6	409	*	255	85.3	44	14.7	299	*	86	78.2	24	21.8	110	NS
		T	19.2	77	74.8	26	25.2	103		55	75.3	18	24.7	73		22	73.3	8	26.7	30	
*D*-*VII*	*rs1334880*	G		337	83.2	68	16.8	405	*	251	85.1	44	14.9	295	*	86	78.2	24	21.8	110	NS
		A	19.0	75	74.3	26	25.7	101		53	74.6	18	25.4	71		22	73.3	8	26.7	30	
*D*-*VIII*	*rs7354899*	C		341	83.4	68	16.6	409	*	255	85.3	44	14.7	299	*	86	78.2	24	21.8	110	NS
		T	19.2	77	74.8	26	25.2	103		55	75.3	18	24.7	73		22	73.3	8	26.7	30	
*D*-*IX*	*rs1334881*	C		337	83.2	68	16.8	405	*	253	85.2	44	14.8	297	*	84	77.8	24	22.2	108	NS
		G	19.3	77	74.8	26	25.2	103		55	75.3	18	24.7	73		22	73.3	8	26.7	30	

^(1)^ The major alleles of *D-1, D-II, D-III, D-IV, D-V, D-VII,* and *D-VII* are wild, but the major alleles of *D-VI* and *D-IX* SNPs are mutant. ^(2)^ MAF, minor allele frequencies; ^(3)^
*p*-value for ANOVA: *, *p* < 0.05; ***, *p* < 0.001; NS; non-significance.

**Table 3 nutrients-16-04423-t003:** The risk variables to increase obesity BMI for 3 years according to *DNAJC6* SNPs.

SNP ^(1)^	Variables	Estimate					
		ß	SE	*p*-Value	OR	Limit (L)	Limit (U)
*D*-*I*	Allele (minor)	0.9512	0.4517	0.0352	2.5890	1.0680	6.2740
	RMR	−0.0024	0.0022	0.001	1.0980	1.010	1.0020
	BW	0.4536	0.0559	<0.0001	1.5740	1.4110	1.7560
	RMR/BW	−0.6026	0.1009	<0.0001	0.547	0.449	0.667
*D*-*VII*	Allele (minor)	0.6209	0.3109	0.0458	1.8610	1.0120	3.4220
	RMR	0.0128	0.0017	<0.0001	1.0130	1.0100	1.0160
	Animal_fat (g) ^(2)^	−0.0122	0.0101	0.2269	0.9880	0.9690	1.0080
	SFA (g) ^(2)^	0.0049	0.0192	0.7968	1.0050	0.9680	1.0430
	Retinol (µg) ^(2)^	−0.0020	0.0010	0.0465	0.9980	0.9960	1.0000
*D*-*VIII*	Allele (minor)	0.6184	0.3112	0.0469	1.8560	1.0080	3.4160
	RMR	0.0129	0.0017	<0.0001	1.0130 *	1.0100	1.0160
	Animal_fat (g) ^(2)^	−0.0119	0.0101	0.2366	0.9880	0.9690	1.0080
	TFA (g) ^(2)^	0.0011	0.0242	0.9653	1.0010	0.9550	1.0500
	SFA(g) ^(2)^	0.0021	0.0609	0.9730	1.0020	0.8890	1.1290
	Retinol (µg) ^(2)^	−0.0020	0.0010	0.0480	0.9980	0.9960	1.0000

^(1)^ *SNP rs*-*number*; *D*-*1 (rs17127601)*, *D*-*VII (rs1334880)*, and *D*-*VIII (rs7354899)*. ^(2)^ All dietary intakes were adjusted according to energy intake in 2009 and 2012, respectively. Abbreviation; RMR (resting metabolic rate), BW (body weight), TFA (total fatty acids), and SFA (saturated fatty acids). *: <0.05.

## Data Availability

Data described in the manuscript will be made available upon request pending application and approval.

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
