# Peer review of "Risk Factors Related to Resting Metabolic Rate-Related DNAJC6 Gene Variation in Children with Overweight/Obesity: 3-Year Panel Study"

_nutrients, 2024, doi:10.3390/nu16244423_

Round 1
Reviewer 1 Report
Comments and Suggestions for Authors
The aim of this study was to examine how DNACJ6 gene variation associated with RMR affects 13 obesity risk factors in children obesity aged 8–9.
Overall, the manuscript has been developed well.
As a reviewer, I would like to draw attention to the unification of the aim of the study, which is different in the abstract and in the Introduction.
It is also necessary to supplement the methodology with a precise description of the methods used - subsection Collection of anthropometric, biochemistry and dietary intakes data.
What measurement protocol was used? What evaluation criteria were used? How was RMR measured? Please add relevant literature.
What was the % of fat (animal, SFA) in the diet? Was the vitamin A met DRI? Were metabolic parameters also taken into account in the classification of obesity? Was the study group metabolically healthy? Please clarify
Reviewer 2 Report
Comments and Suggestions for Authors
The manuscript by Shin et al. entitled “Risk factors related to RMR-related DNAJC6 gene variation in overweight/obesity children; A 3-Year Panel Study” investigated how DNACJ6 gene variation associated with RMR affects obesity risk factors in children obesity aged 8–9.
The results suggest that the RMR/BW ratio and dietary fat/retinol intake should be considered in DNACJ6 gene-based precision medicine approaches for pediatric obesity prevention, particularly for boys.
I think the topic is very interesting and the approach used by the authors is correct.
There are some aspects that should be better explained and reviewed:
1) 1) L. 94-95: Did you collect information on what the children did in the three-year interval?
2) L. 124-125: It is not clear to me whether the SNPs were commercially available or designed by the authors. Also, are the genes evaluated of local interest or have they been associated with obesity in other populations?
3) 3) In general, authors should carefully check the entire text to eliminate the various errors present.
4) L. 184-185: Why did the intake quality vary so much?
5) L. 192-195: this is a conclusion/consideration, not a result
6) Fig. 3A: why not indicate the exact p?
7) Fig. 3B: why is there no DS or SE?
8) Discussion: other indicators, such as RMR/BW. However, this report is not exhaustive: body composition and especially the distribution of fat mass in visceral and subcutaneous are more precise indicators. The authors should discuss this.
